# *N*-Acetyl-l-phenylalanine Racemization during TBTU Amidation: An In-Depth Study for the Synthesis of Anti-Inflammatory 2-(*N*-Acetyl)-l-phenylalanylamido-2-deoxy-d-glucose (NAPA)

**DOI:** 10.3390/molecules28020581

**Published:** 2023-01-06

**Authors:** Elisa Sturabotti, Fabrizio Vetica, Giorgia Toscano, Andrea Calcaterra, Andrea Martinelli, Luisa Maria Migneco, Francesca Leonelli

**Affiliations:** 1Department of Chemistry, Sapienza University of Rome, P.le Aldo Moro 5, 00185 Rome, Italy; 2Department of Chemistry and Technology of Drugs, Sapienza University of Rome, P.le Aldo Moro 5, 00185 Rome, Italy

**Keywords:** racemization, *N*-acetyl amino acids, coupling, TBTU, pyridine

## Abstract

A thorough study on the amidation conditions of *N*-acetyl-l-phenylalanine using TBTU and various bases is reported for the synthesis of 2-(*N*-acetyl)-l-phenylalanylamido-2-deoxy-d-glucose (NAPA), a promising drug for the treatment of joints diseases. TBTU-mediated diastereoselective amidation reaction with 1,3,4,6-tetra-*O*-acetyl-β-d-glucosamine always gave racemization of *N*-acetyl-l-phenylalanine. The stereochemical retention under amidation conditions was studied in detail in the presence of difference bases and via other control experiments, evidencing the possibility to reduce racemization using pyridine as base.

## 1. Introduction

Amide bonds are mostly obtained through the condensation of carboxylic acids and amines, in a two-step reaction which starts with the activation of the carboxylic compound by a coupling agent/tertiary base pair and proceeds with the attack of the nucleophilic amine. Molecules that bear amidic moieties in their structure are often characterized by stereocenters that belong to carboxylic acid, amine, or both. The chirality of those sites should be preserved to maintain the properties of the substrates unaltered, conferring peculiar features to the synthetized molecules, especially in the case of bioactive compounds. Nevertheless, the condensation of *N*-protected chiral amino acids is known to be accompanied by substantial racemization at the α proton [1], resulting in the loss of stereocenter configuration in the target amides. Several causes contribute to the racemization of chiral substrates, but the phenomenon usually takes place on the *N*-protected amino acid/coupling agent reactive intermediate generated during the activation of the amino acid (*vide infra*) [2]. In fact, the latter could self-cyclize into an azlactone, characterized by an acid α-proton (pK_a_ ≈ 9), which is responsible for the abundant epimerization/racemization generally occurring during the amidation reaction [3]. Among other strategies, to reduce or avoid racemization, an in situ activation strategy or a fast activation process are generally proposed to effectively shorten the existence period of the activated amino acid species [4].

During the past few years, the formerly predominant carbodiimide and active ester coupling techniques have been replaced with uronium salts based upon 1-hydroxybenzotriazole (HOBt) or 7-aza-1-hydroxybenzotriazole (HABt) [5], namely HBTU, HATU, TBTU or TATU [6]. Uronium-based coupling agents have gained much attention since they are able to rapidly activate amino acids in polar solvents with low racemization and side reactions [7], even in the case of difficult sterically hindered condensations [8]. Those features are mostly related to the HOBt and HABt component, since it interferes in the activation of the amino acid, playing a significant role in reducing racemization [9]. 

In the frame of biological active compound synthesis [10,11,12], we focused on the preparation of 2-(*N*-acetyl)-l-phenylalanylamido-2-deoxy-d-glucose (NAPA) [13], an *N*-acetyl-l-phenylalanine bioactive derivative of glucosamine hydrochloride (Glu) that has shown interesting chondroprotective action in the field of joint degenerative diseases [14].

Nowadays, glucosamine, alone or in addition to chondroitin sulfate, is commonly used for the oral treatment of osteoarthritis (OA) [15], a degenerative joint disease that causes articular alterations, local inflammations, and joint effusions. Glucosamine seems to have a fundamental role in the synthesis of glycosaminoglycans, which are one of the main constituents of cartilage [16], and its usage on animals affected by OA would seem to enrich their cartilage tissue [17]. Since OA and rheumatoid arthritis afflict many patients all around the world (it is estimated that more than 25% of the over 50-year-olds suffer from OA and more than 80% of those over 65 suffer from some form of degenerative joint disease) [18], the discovery of novel active therapeutics treatments is of pivotal importance. With respect to NAPA, some in vitro investigations have highlighted that its intra-articular injections on pathogenic animals are associated with a more homogeneous chondrocyte cellularity, absence of fissures and fragmentation in tissues and a more solid and smooth appearance of the matrix [19]. Furthermore, NAPA has been revealed to have anti-inflammatory activity on chondrocytes, counteracting the local cellular inflammation and contrasting specific cytokines’ production [20]. Therefore, the molecule could represent a promising drug for the treatment of joint diseases [21].

From a structural point of view, NAPA (**2b** in Figure 1) is composed of a D-glucosamine unit in which the NH_2_-C(2) position is amidated with *N*-acetyl-l-phenylalanine (*N*-Ac-l-phe).

Considering NAPA’s molecular structure (**2b**), we envisioned its synthesis relying on a direct amidation reaction between *N*-acetyl-l-phenylalanine and 1,3,4,6-tetra-*O*-acetyl-glucosamine (**1**), which could be easily prepared following a reported procedure [22]. The condensation step, although based on a simple reaction, represents the most delicate point in the process because of the ubiquitous and recurring tendency of *N*-acetyl protected amino acids to give azlactones, i.e., to racemize, when activated by a coupling agent under a basic environment [1]. The intermediate **L-2a** could be converted to NAPA via direct *O*-deacetylation reaction. 

Preliminary trials of amidation between *N*-Ac-l-phe and **1** to obtain NAPA with conventional coupling reactants, such as IBCF [13] or EDC hydrochloride, lead to substantial levels of amino acid racemization (unpublished results); hence, we envisioned the possibility of using HOBt uronium-salt derivatives due to their advantageous properties over other coupling reactants. 

Within this context, 2-(1H-Benzotriazole-1-yl)-1,1,3,3-tetramethylaminiumtetrafluoroborate (TBTU) was chosen as a condensing agent in our retrosynthetic design for the diastereoselective synthesis of NAPA because of the need to maintain the L center of the peptidyl residue unaltered in the target. This research presents the optimization of the conditions for the diastereoselective synthesis of a bioactive derivative of glucosamine, with a detailed discussion of the amidation reaction mechanism.

## 2. Results and Discussions

As mentioned above, following unsuccessful results obtained with classical condensation agents, our initial attempts to synthetize **L-2a** were made using TBTU along with *N,N*-diisopropylethylamine (DIPEA, pK_a_ ≈ 11), according to standard known amidation procedures reported for uronium salt, in which two or more equivalents of bases were required [23,24]. The hypothesized mechanism for the amidation of *N*-Ac-l-phe using TBTU is reported in Figure 2. After deprotonation of the amino acid by a suitable base, activation starts with the attack of the carboxylate on the imide carbonyl carbon of TBTU, thus releasing the benzotriazole *N*-oxide anion. The latter reacts with the electrophilic carbonyl group to give an active ester, forming tetramethyl urea as a byproduct. The aminolysis of the HOBt ester gives the desired product [25]. The role of the HOBt unit in racemization suppression is mainly based on its reaction with the activated amino acid to give a more reactive ester, enhancing the possibility of the amine to attack [26]. 

The coupling reaction of *N*-Ac-l-phe with **1** using TBTU/DIPEA is reported in Figure 3, while the experimental conditions are listed in Table 1. Under these experimental conditions, the racemization of the *N*-Ac-l-phe moiety was always observed, hence leading to the generation of the two epimeric products **L-2a** and **D-2a**.

With an excess of DIPEA, amidation of *N*-acetyl-l-phenylalanine always resulted in a mixture of diastereoisomers, with **D-2a** as a major product (entries 1 and 2). Time shortening did not affect the reaction outcome positively. Then, the DIPEA amount was lowered to one equivalent (amino acid:base = 1:1), whereas the chirality loss of the amino acid is often ascribed to a base-promoted deprotonation of the activated amino acid [27]. An equimolar amount of DIPEA led, again, to a mixture of **2a** diastereoisomers within 24 or 3 h (entries 3 and 4). Those results were unexpected especially if a complete “consumption” of the base during activation of the amino acid is assumed (Figure 2). The temperature decrease still gave a high degree of racemization, but in this case, **L-2a** was the major diastereoisomer (entry 5).

In light of the above, it could be inferred that the presence of a strong base, such as DIPEA, has a pivotal role in the inversion of chirality, independently from its amount. Hence, further experiments have been carried out, keeping in mind the importance of base choice in the coupling of *N*-Ac-l-phe. As previously mentioned, the racemization of amino acids occurs during the activation step, particularly in the cases of *N*-acetyl or *N*-benzoyl amino acids [28,29], and it is ascribed to the cyclization of the activated intermediate to give a heterocyclic azlactone (**L-3** in Figure 4). An intramolecular reaction between the oxygen at the acetyl *N*-protector group and the electrophilic site in C-1 promotes its formation [30]. The α-hydrogen of the azlactone is characterized by moderated acidity (pK_a_ ≈ 9), attributed to the aromatic character of the corresponding enolate tautomer [31].

The intermediate **L-3** could rapidly rearrange into the epimeric/racemic compound **3** by α-deprotonation or the keto-enol tautomerism, becoming itself the intermediate that undergoes the nucleophile attack of the amine group. As a result, the amide **2a** could be obtained as a mixture of diastereoisomers with stereochemical properties that depend upon the reactivity of all the species involved. To further support our proposed mechanistic pathway, we needed to exclude the possibility of racemization of the generated product during the reaction course. Thus, we probed pure compound **L-2a** and 50:50 or 30:70 mixtures of **L-2a**:**D-2a** under standard reaction conditions (amide:base:TBTU = 1:2:1, 24 h, r.t.). Pleasingly, a complete stereochemical retention of the products was observed, evidencing that the racemization occurs necessarily during the activation process.

Considering the abovementioned acidity of the azlactone (pK_a_ ≈ 9), DIPEA (pK_a_ ≈ 11) in excess could reasonably cause the formation of both **L-2a** and **D-2a** (Table 1, entries 1 and 2). The crude product is already enriched in **D-2a** after three hours of reaction, meaning that formation of **L-3** is fast, and its racemization is promoted by the base excess. Surprisingly, when base and amino acid are equimolar, inversion of chirality takes place in 3 h (Table 1, entries 3 and 4) and **D-2a** is still the most abundant product, even at a low temperature (Table 1, entry 5). Beyond the mechanisms that induce racemization, a major reactivity of D-activated enantiomer could be assumed, and further experiments were carried out to prove this aspect. **D-2a** was synthetized from *N*-acetyl-d-phenylalanine (*N*-Ac-d-phe) in the presence of one or two equivalents of DIPEA (entries from 6 to 9 in Table 1). Amide **D-2a** is always obtained as a major product in good yields. Racemization of *N*-Ac-d-phe with an equimolar base could be considered completed after 3 h, matching the results of long and short reactions. Lastly, a good effect emerges from the temperature lowering because more than 90% of the product is represented by **D-2a** (entry 9).

It is now obvious that stereochemistry retention was better attained for *N*-Ac-d-phe condensation, thereby remarking the major reactivity of d-activated isomer upon tetra-*O*-acetyl-β-d-glucosamine, regardless of the base amount. In fact, if the azlactone formation and racemization are fast, the stereochemical outcome of the coupling will be determined by the relative rate of reaction of oxazolone stereoisomers with the chiral nucleophile component [32]. This explanation is consistent with our results since the diastereoselectivity in the coupling reaction seems to be guided by the major reactivity of d-isomer toward **1**, both when the d-enantiomer is the starting reagent (Table 1, entries 6–9) and when it is somehow formed in situ (Table 1, entries 1–5).

Stereochemistry retention in the TBTU-mediated coupling was also tested in the case of the *N*-carbamate protector group, which is known to easily endure racemization [33]. In this regard, L-phenylalanine bearing a *N*-carbamate protecting group (*N*-(Carbobenzyloxy)-l-phenylalanine, *N*-Cbz-l-phe) was coupled to **1** following entries 1 and 3 in Table 1 (Figure 5 and Table 2). 

Synthesis of **L-4c** occurs in high yields and without the detection of epimerization, both for entries 1-Z or 2-Z, after 24 h. These results could be explained considering the higher stability of *N*-Cbz protected amino acid towards racemization.

Coupling of *N*-Ac-l-phe with the amino sugar **1** was also investigated using 1-propanephosphonic anhydride (T3P, mechanism in Figure 6), since the formation of several T3P-mediated amide bonds can be found in the literature on epimerization-prone substrates [34,35]. For instance, *N*-Fmoc-l-amino acids, including *N*-Fmoc-l-phe, were coupled to 2,3,4,6-tetra-*O*-acetyl-β-d-glucopyranosyl-1-amine, an isomer of **1**, via T3P and TBTU using a molar excess of DIPEA. In their experiments, the amidation of *N*-Fmoc-l-phe with 2,3,4,6-tetra-*O*-acetyl-β-d-glucopyranosyl-1-amine using T3P/DIPEA gave an optically pure product, unlike that obtained with TBTU/DIPEA coupling system [36].

**L-2a** was synthetized using T3P/DIPEA (2 eq.) as a coupling system to compare and evaluate the T3P effect on preventing the epimerization of *N*-Ac-l-phe. For the sake of comparison, *N*-Cbz-l-phe was also reacted with **1**. A list of T3P-mediated reactions is reported below in Table 3.

Unfortunately, **2a** was obtained as a mixture of diastereoisomers, with an L:D ratio of 34:66, while **L-4c** was isolated as an optically pure compound (entry 2-T3P). Again, racemization of *N*-Ac-l-phe was obtained proving the high tendency of the L amino acid towards racemization, regardless of the type of coupling agent used for the activation.

Although these results offer an efficient approach to form the desired amide bond suppressing racemization using a *N*-carbamate protection, with our target molecule possessing an *N*-acetyl function, this method would necessitate a further deprotection of the Cbz group, followed by an *N*-acetylation step to restore the functionality desired in the target drug. From a synthetic point of view, the avoidance of protection/deprotection steps is always preferred when possible. Thus, we decided to continue our investigation on *N*-Ac-l-phe, searching for a way to suppress racemization.

If the action of excess DIPEA on the oxazolone is clear, the results obtained for TBTU with one equivalent of it are not straightforward. Therefore, the reaction of N-Ac-l-phe was thoroughly studied in order to understand and prevent racemization. Although a variety of excellent examples are proposed in the literature for *N*-Fmoc, *N*-Cbz, and *N*-Boc protected amino acids [37,38], only a few studies are related to *N*-acetyl protected amino acids, especially in the presence of chiral amines; thus, we investigated in depth the effect of the strength of the stoichiometric base on the chirality retention. A variety of bases have been explored with TBTU, keeping a 1:1 ratio between *N*-Ac-l-phe and the base (Table 4).

Analyzing the results reported in Table 4, it can be noticed that epimerization at the L-stereocenter was a recurring phenomenon, but L–D inversion showed a base-related trend. The L stereocenter in the amino acid is better preserved with equimolar *N,N*-dimethylaniline (DMA), pyridine (Py) or the more hindered 2,6-lutidine (Lut) (Table 4, entries 3–9, pK_a_ < 7), while 4-Dimethylaminopyridine (DMAP), *N,N*-diisopropylethylamine (DIPEA), and 1,8-diazabiciclo[5.4.0]undec-7-ene (DBU) promote the predominant formation of **D-2a** (Table 4, entries 10–12, pK_a_ > 9). The best diastereoisomeric ratio was obtained with DMA in 3 h (97:3), but, interestingly, the reaction conducted with TBTU and equimolar Py gave a L:D ratio or 93:7 with an optimum diastereoisomeric abundance (40%). Along with yield, **D-2a** increases gradually with time when pyridine is used (entry 5), which is a sign of a slower epimerization. It is worth noting that the temperature decrease (entries 6 and 7) and the increase in pyridine quantity (entry 8) do not affect the coupling outcome much, leading in many cases to **L-2a** as the major product, differently from what was previously observed for DIPEA.

When the reaction is carried out without a base (entries 1 and 2, Table 4), no traces of product are observed within 3 h (TLC analysis), and the yield after 24 h is low (38%). A small amount of **D-2a** is still present in the crude products. Thus, from the last results, it is clear that the base is crucial for the deprotonation, which led to the subsequent amino acid activation, and hence for the amidation reaction. Stronger bases will efficiently deprotonate the acid, and the resulting carboxylate will rapidly interact with the TBTU to give an activated amino acid intermediate (Figure 2). Therefore, we can assume that the more the amino acid is quickly deprotonated from a strong base, the more it will be activated by TBTU and, regrettably, rearranged into the azlactone **L-3**, always in equilibrium with **3** through the keto-enol tautomerism (Figure 4).

For this reason, equimolar DMAP, DIPEA, or DBU promote the formation of TBTU-amino acid intermediate giving, at the same time, its racemization. As mentioned above, the diastereoselectivity of amine **1** toward the d-isomer would guide the reaction, causing the preferential formation of **D-2a**. DMA, Py, and Lut afford a lower degree of activation, limiting the formation of **L-3** and the overall racemization but, also, the reaction yield. To study the racemization avoiding the stereoselectivity of d-glucosamine, aniline was chosen as nucleophile (Figure 7), and it was reacted with pure l, pure d-phe, or a 1:1 mixture of L and D amino acid, using equimolar DIPEA or pyridine. The enantiomer ratios of the final chiral compound **5** were analyzed via HPLC using a chiral stationary phase. According to the data presented in Table 5, independently from the type of nucleophile, the TBTU/DIPEA-mediated coupling induces racemization, while the TBTU/pyridine system completely preserves the L-center in the product, corroborating the idea that the phenomenon is strongly associated with the activation step of the two *N*-acetyl-phenylalanines (Table 5).

Comparing the R^E^ and yields, it is clear that to stress the formation of the amino acid-TBTU, active intermediate strong bases are necessary, anyway leading to the formation of a racemization-susceptible azlactone. Since the achiral aniline has no influence on the diastereoselectivity of the process, enantiomers **L-5** and **D-5** are obtained in similar amounts. In contrast, the inefficiency of the amino acid activation with a weak base, such as pyridine, reflects in the poor conversion of aniline into pure **L-5**.

## 3. Materials and Methods

### 3.1. General Information

d-glucosamine HCl, TBTU, T3P, dry DMF, tertiary bases, solvents, and amino acids were purchased from Sigma Aldrich. All chemicals were used without any purification. Before amidation, the optical purity of the L and D amino acids (*N*-Ac-phe and *N*-Cbz-phe) was confirmed via polarimeter measurements at 25 °C.

### 3.2. Synthesis of **L-2a** Using TBTU or T3P

*N*-Ac-l-phe (1 mmol) and a selected tertiary base (1 or 2 mmol) were dissolved in dry DMF (2.5 mL) under argon atmosphere. Then, TBTU or T3P was added (1 mmol) and mixed for 5 min (activation step). After that time, **1** (1 mmol) in dry DMF (0.2 mL) was dripped to the activated amino acid, and the mixture was stirred in r.t. Reaction was quenched with water (10 mL), and the white precipitate was separated by filtration and vacuum-dried. The amount of **L-2a** and **D-2a** in the precipitate was determined by the ratio between the anomeric protons in ^1^H NMR spectra. Pure **L-2a** was isolated by crystallization in acetone (see Appendix A).

**L-2a**: [α]_D_^25^ = −14 (c = 1.0 in CHCl_3_). ^1^H NMR for **L-2a** (400 MHz, CDCl_3_) δ 7.40–7.00 (m, 5H, ArPhe), 6.73 and 6.04 (2d, 2H, 2 × NH) 5.86 (d, *J* = 8.8 Hz, 1H, H-1β-l), 5.33 (t, *J* = 9.9 Hz, 1H, H-3), 5.08 (t, *J* = 9.5 Hz, 1H, H-4), 4.53 (m, 1H, CHPhe), 4.32–3.96 (m, 3H, H-2,6,6′), 3.85 (m, 1H, H-5), 3.11 (dd, *J* = 14.0, 6.3 Hz, 1H, CH_2_Phe), 2.85 (dd, *J* = 14.0, 7.2 Hz, 1H, CH_2_Phe), 2.07, 2.05, 2.00, 1.96 and 1.91 (5s, 15H, CH_3_).

^13^C NMR for **L-2a** (101 MHz, CDCl_3_): 171.65, 171.01, 170.82, 170.65, 169.52, 169.43, 136.31, 129.21, 128.85, 127.22, 92.23 (C-1β-l), 72.83, 71.68, 68.16, 61.82, 54.66, 53.79, 37.21, 23.07, 21.01, 20.85, 20.75, 20.71.

ATR-FTIR (powder): main peaks at 3300, 1739, 1667, 1649, 1365, 1200, 1035, 700 cm^−1^. ESI-MS for **L-2a**: +Q1: Calculated for C_25_H_32_N_2_O_11_^+^ [M + H]^+^: 537.2; measured m/e: 537.5.

### 3.3. Synthesis of **D-2a**

**D-2a** was synthetized using the same protocol adopted for **L-2a** but replacing *N*-Ac-l-phe with *N*-Ac-d-phe. Pure **D-2a** was purified by crystallization in ethyl acetate.

**D-2a**: [α]^D^_25_ = +37 (c = 1.0 in CHCl_3_). ^1^H NMR for **D-2a** (400 MHz, CDCl_3_): δ 7.40–7.00 (m, 5H, ArPhe), 6.90 and 6.40 (2d, 2H, 2 × NH), 5.76 (d, *J* = 8.8 Hz, 1H, H-1β-d), 5.28 (t, *J* = 9.9 Hz, 1H, H-3), 5.13 (t, *J* = 9.5 Hz, 1H, H-4), 4.59 (m, 1H, CHPhe), 4.33–4.07 (m, 3H, H-2,6,6′), 3.87 (m, 1H, H-5), 3.07 (dd, *J* = 14.1, 6.3 Hz, 1H, CH_2_Phe), 2.86 (dd, *J* = 14.1, 8.1 Hz, 1H, CH_2_Phe), 2.07, 2.05, 2.03, 2.00 and 1.86 (5s, 15H, CH_3_).

^13^C NMR for **D-2a** (101 MHz, CDCl_3_): δ 171.95, 171.30, 170.77, 170.57, 169.46, 169.38, 136.49, 129.06, 128.79, 127.16, 92.02 (C-1β-d), 72.94, 72.88, 68.11, 61.91, 54.77, 52.94, 37.46, 22.98, 20.99, 20.83, 20.64.

### 3.4. Synthesis of **L-4c**

**L-4c** was synthetized using the same protocol adopted for **L-2a** but replacing *N*-Ac-l-phe with *N*-Cbz-l-phe.

^1^H NMR for **L-4c** (400 MHz, CDCl_3_) δ 7.44–7.04 (m, 10H, 2 × Ar), 6.49 (d, 1H, NH), 5.73 (d, *J* = 8.7 Hz, 1H, H-1β-l), 5.21 (t, 1H), 5.12 (t, 1H), 5.03 (s, 2H, CH_2_Cbz) 4.50–4.50 (m, 4H), 3.80 (d, 1H), 3.18–2.78 (2 × dd, 2H, CH_2_Phe), 2.08, 2.06, 2.02 and 1.97 (4s, 12H, CH_3_).

### 3.5. Synthesis of **5**

Coupling of aniline with *N*-Ac-l-phe and *N*-Ac-d-phe using TBTU: pure amino acids or the selected mixture were reacted with aniline following the coupling procedure described for **L-2a**. Enantiomers of **5** were separated using chiral HPLC (Chiralpak, IA Hexane—iPrOH:95:5, 1 mL/min).

^1^H NMR for **5**: (400 MHz, CDCl3): δ 8.07 (s, 1H, NH), 7.40–6.90 (m, 10H, Ar), 6.41 (~d, 1H, NH), 4.83 (q, 1H, CHPhe), 3.13 (m, 2H, CH_2_Phe), 2.01 (s, 3H, CH_3_Phe) [40].

## 4. Conclusions

In conclusion, the TBTU coupling agent was tested to obtain the optically pure bioactive compound NAPA from the reaction of *N*-acetyl-l-phenylalanine with 1,3,4,6-tetra-*O*-acetyl-β-d-glucosamine, maintaining the *N*-protecting group of the amino acid. Due to the consistent loss of the amino acid stereochemical purity, several condensation procedures have been proven and discussed. In particular, TBTU was reacted with different bases, following the observation that the degree of racemization was related to the strength of base used. A reasonable description of the underlying mechanism was proposed, considering the role of the base involved in the activation of the L-amino acid and the diastereoselectivity induced by the chiral glucosamine. Our results suggested that the standard procedures for the activation of amino acids with TBTU, which require a high concentration of a strong base, are reasonably inefficient for *N*-Ac-l-phe because of the formation of an azlactone intermediate. We demonstrated that in the coupling with 1,3,4,6-tetra-*O*-acetyl-β-d-glucosamine or aniline, the L-center was better preserved using one equivalent of pyridine during the activation step, while bases with a higher pK_a_, such as DMAP, DIPEA or DBU, sharply altered the stereocenter purity both when equimolar to the amino acid and in excess. This research, which starts from our interest in the synthesis of the bioactive molecule NAPA, represents a strict and comparative study on the conditions that lead to the racemization of *N*-acetyl-l-phenylalanine during a uronium-mediated amidation. Although with medium yields, mainly limited by the diastereoselectivity of the 1,3,4,6-tetra-*O*-acetyl-β-d-glucosamine toward the L-amino acid, we evidenced that the use of a stoichiometric amount of a weak base has a beneficial effect on the chirality retention of *N*-acetyl-l-phenylalanine during coupling.

The TBTU/pyridine mediated reaction afforded the desired anti-inflammatory compound NAPA with excellent stereochemical retention, and it could potentially be extended to all those TBTU-mediated couplings involving racemizable substrates, for example, the *N*-acetylated amino acids, where chirality retention is required.

## Figures and Tables

**Figure 1 molecules-28-00581-f001:**
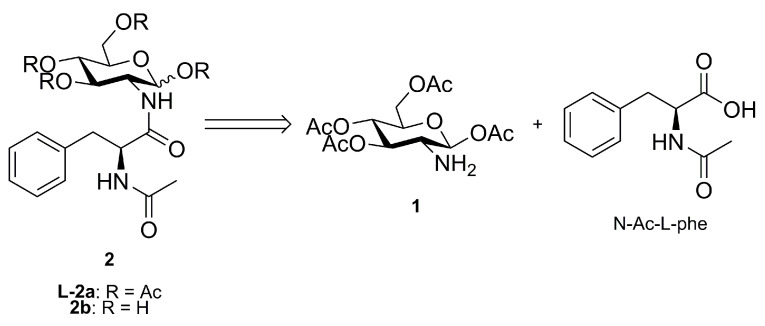
Retrosynthesis for NAPA (**2b**) or its tetra-*O*-acetylated precursor (**L-2a**) from 1,3,4,6-tetra-*O*-acetyl-β-d-glucosamine (**1**) and *N*-acetyl-l-phenylalanine (*N*-Ac-l-phe).

**Figure 2 molecules-28-00581-f002:**
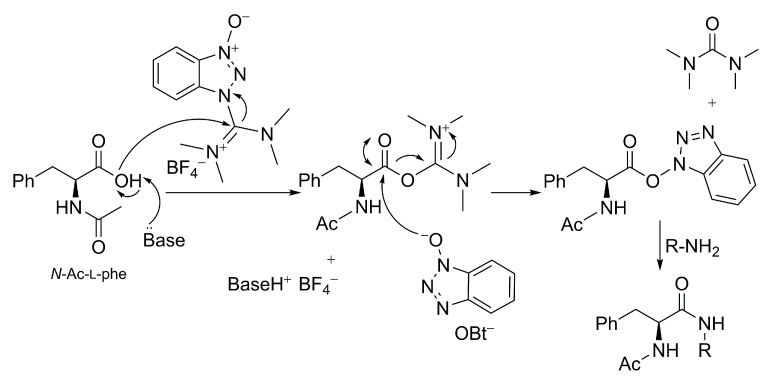
Hypothesized mechanism of *N*-Ac-l-phe activation and amidation using TBTU and a tertiary base.

**Figure 3 molecules-28-00581-f003:**
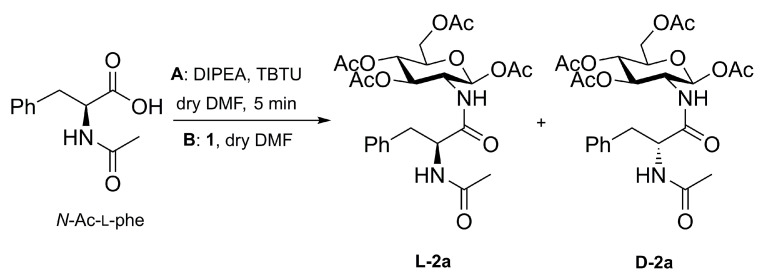
Synthesis of diastereoisomers **L-2a** and **D-2a** starting from enantiopure *N*-acetyl-l-phenylalanine. **A**: activation step; **B**: amidation with **1**.

**Figure 4 molecules-28-00581-f004:**
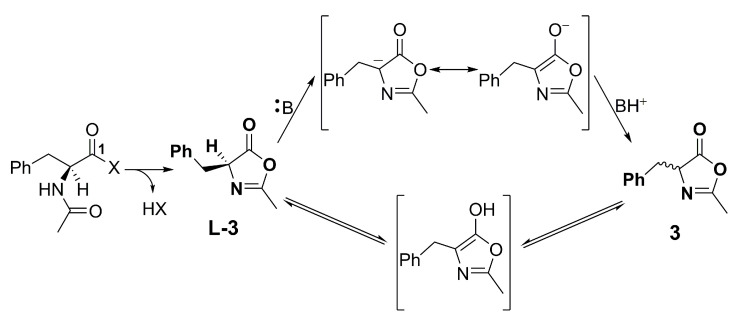
Theorized azlactone formation during the activation of *N*-Ac-l-phe with TBTU. X is the activating–leaving group.

**Figure 5 molecules-28-00581-f005:**
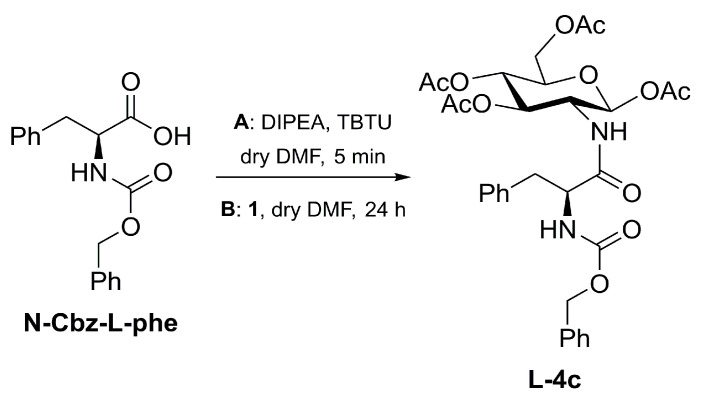
Synthesis of **L-4c** by means of TBTU/DIPEA coupling. DIPEA is used in excess with respect to the amino acid in order to evaluate a possible racemization.

**Figure 6 molecules-28-00581-f006:**
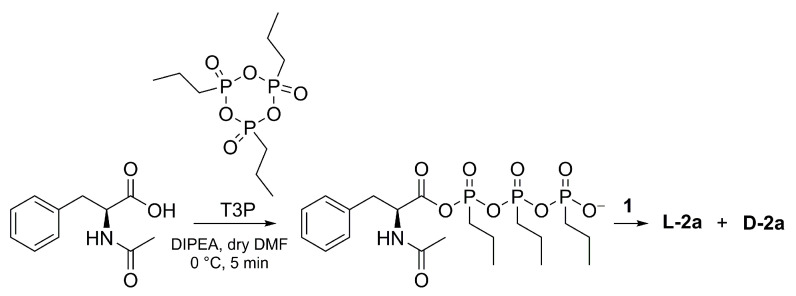
Synthesis of **L-2a** in coupling mediated by T3P. The activation of N-Ac-l-phe starts with the deprotonation of the carboxylic acid and continues with the attack of electrophile cyclic phosphonic anhydride.

**Figure 7 molecules-28-00581-f007:**
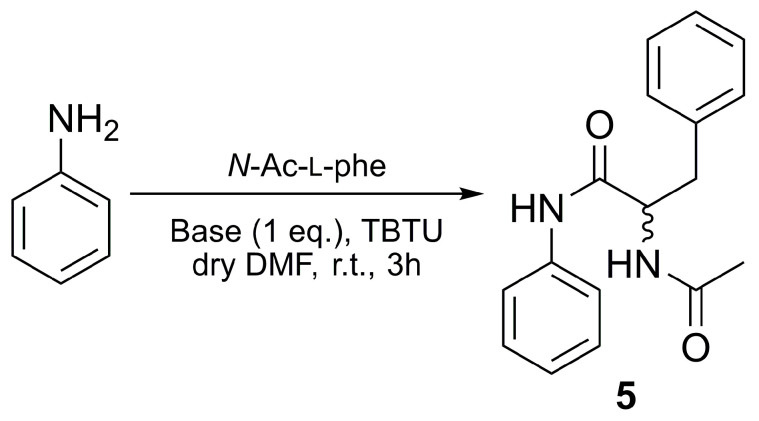
Synthesis of 2-acetamido-N,3-diphenylpropanamide (**5**) from *N*-Ac-phe and aniline. The coupling is mediated by TBTU.

**Table 1 molecules-28-00581-t001:** Experimental parameters for the coupling of **1** with *N*-Ac-l/d-phe mediated by TBTU/DIPEA. T_R_ = reaction temperature, t_R_ = reaction time, R^D^ = diastereoisomeric ratio (**L-2a**:**D-2a**). Yield is referred to the sum of the diastereoisomer ^1^.

Entry	Amino Acid Configuration	DIPEA (eq)	T_R_ (°C)	t_R_ (h)	Yield (%)	R^D^ (L-2a:D-2a)
1	L	2	r.t.	24	87	31:69
2		2		3	82	34:66
3		1		24	78	24:76
4		1		3	71	29:71
5		1	−10	3	49	65:35
6	D	2	r.t.	24	93	35:65
7		1		24	93	24:76
8		1		3	82	26:78
9		1	−10	3	88	8:92

^1^ Reactions were carried out in triplicate.

**Table 2 molecules-28-00581-t002:** Specifics for *N*-Cbz-l-phe and **1** coupling in the presence of TBTU/DIPEA. T_R_ = reaction temperature, t_R_ = reaction time, R^D^ = diastereoisomeric ratio (**L-4c**:**D-4c**).

Entry	DIPEA (eq.)	T_R_ (°C)	t_R_ (h)	Yield (%)	R^D^ (L-4c:D-4c)
1-Z	1	r.t.	24	80	100:0
2-Z	2	r.t.	24	85	100:0

**Table 3 molecules-28-00581-t003:** T3P coupling specifics for the synthesis of **L-2a** or **L-4c**. T_R_ = reaction temperature, t_R_ = reaction time, R^D^ = **L**:**D** ratio.

Entry	Amino Acid	DIPEA (eq.)	T_R_ (°C)	t_R_ (h)	Yield (%)	R^D^ (L:D)
1-T3P	*N*-Ac-l-phe	2	r.t.	24	84	34:66
2-T3P	*N*-Cbz-l-phe	2	r.t.	24	87	100:0

**Table 4 molecules-28-00581-t004:** Experimental parameters for the coupling of **1** with *N*-Ac-l-phe mediated by TBTU and different bases. Eq. = base equivalents, T_R_ = reaction temperature, t_R_ = reaction time, R^D^ = diastereoisomeric ratio (**L-2a**:**D-2a**) ^1^.

Entry	Base ^2^	Eq.	T_R_ (°C)	t_R_ (h)	Yield (%)	R^D^ (L-2a:D-2a)	L (%)	D (%)
1	-	-	r.t.	3	-	-	-	-
2	-	-	r.t.	24	38	80:20	30	8
3	DMA	1	r.t.	3	30	97:3	29	1
4	Py	1	r.t.	3	43	93:7	40	3
5		1	r.t.	5	50	87:13	44	6
6		1	0	5	16	100:00	16	0
7		1	5	15	43	87:13	37	6
8		2	r.t.	3	40	93:7	37	3
9	Lut	1	r.t.	3	55	83:17	46	9
10	DMAP	1	r.t.	3	71	35:65	25	46
11	DIPEA	1	r.t.	3	71	29:71	21	50
12	DBU	1	r.t.	3	64	40:60	26	38

^1^ Reactions were carried out in triplicate. ^2^ pK_a_ values of the bases are calculated in water [39].

**Table 5 molecules-28-00581-t005:** Preparation of amide **L-5** or **D-5** from optically pure amino acids or their 1:1 mixture. Coupling is carried out using TBTU with 1 eq. of *N,N*-diisopropylethylamine or pyridine. t_R_ = reaction time.

Entry	Amino Acid Conf.	Base	Eq.	t_R_ (h)	Yield (%)	R^E^ (L-5:D-5)
1	L	DIPEA	1	3	60	56:44
2	D		1		59	52:48
3	L:D = 1:1		1		66	51:49
4	L	Py	1		15	98:2

## Data Availability

Not applicable.

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
