# Peer review of "N-Acetyl-l-phenylalanine Racemization during TBTU Amidation: An In-Depth Study for the Synthesis of Anti-Inflammatory 2-(N-Acetyl)-l-phenylalanylamido-2-deoxy-d-glucose (NAPA)"

_molecules, 2023, doi:10.3390/molecules28020581_

Round 1

Reviewer 1 Report

Francesca Leonelli and co-authors reported the studies on N-acetyl-L-phenylalanine racemization during TBTU amidation. 2-(N-acetyl)-L-phenylalanylamido-2-deoxy-D- 17 glucose (NAPA) – a promising drug - was synthesized from the amino acid and glucosamine. The racemization of acetylphenyalanine is an undesirable process, and the authors have made attempts to carry out the reaction in such a way as to avoid this phenomenon. For this, various reaction conditions, bases, ratios were varied, protective groups were introduced, which led to positive results. On the one hand, the described procedures are well known and even trivial. Also, the data and suggestions obtained can be considered expected for the most part. However, I read the manuscript with interest because the design of the research, the order of presentation, and the quality of the writing were of a very high standard. I think, that the presented results will be of interest to a wide scope of readers, as they will lead to a better understanding of the described phenomena and principles. Thus, I am in favor of publication these data in Molecules journal after a minor revision.

- Were there any attempts by the authors to register any intermediates using the HRMS? The described products contain a nitrogen atom, and these compounds should be visible in the spectra.

- the sentence “Following our interest in the synthesis of biological active compounds [10-12]” should be removed from the text or re-written, since previous studies relate to other topics.

typos to fix

-          Figure 6. . Synthesis

-          T3P/DIPEA gave a optically

Reviewer 2 Report

The authors have examined in detail the racemization of N-acetyl-L-phenylalanine in the condensation reaction with amines using TBTU, using NAPA, which has anti-inflammatory properties, as an example. Although racemization in the condensation reaction of amino acids is an important event that is of interest to many synthetic organic chemists, few papers have discussed it in detail, especially the conditions. From this point of view, I think this paper is suitable for publication in Molecules. However, the following points should also be considered.

Figure 5.and Table 2.Is 100% of the stereochemistry retained when N-Cbz-D-phe is used as the starting material?

If N-Cbz-L-phe is used, is the molecule corresponding to L-3 undetectable?

It would be good to have a description of these.

Figure 6..Figure 7.. Two periods.

Reviewer 3 Report

I think this article could be published after major revision. While experiments itselves seem fine to me, I’m not quite convinced/agreed with some of the conclusions made by authors based upon these presented data. I think the authors should run some additional experiments to strongly support their statements or at least should provide extra reasons/explanations why they think that their data clearly and unambiguously supports their initial way of thoughts.    1)      Based on results in the Table 1 authors came to a conclusion that racemization degree depends on DIPEA amounts, but to me it doesn’t seem that clear. In both cases (entry 1-2 and 3-4) the ratio of L-2a : D-2a is about 1 to 2-2.5. Such dispersion could be easily statistically insignificant; I would suggest the authors run several repetition experiments in order to support their statement with statistical analysis.   2)      Extra dot in the title Figure 6.   3)      Again, in my opinion without clear evidence that amine excess promotes the racemization (see for example entry 8 in Table 4) it may make sense to rerun experiments in Table 4 with 2 or more equivalents of base. I think it either will supports the authors’ reasoning about the role of the excess of tertiary amine or unproved it by leading to better yields while keeping at bay L/D – isomers ratio. Regarding the fact that isomerization happens even without any base upon the prolonged reaction time (entry 2) it could be that excess of base plus shorten reaction time plus low temperature would result both in decent yield and stereochemical purity (optimization studies)   4)      I don’t quite understand/agree with authors statement that “…..once the amino acid reacts with the base (Figure 2), we can assume that the more it is activated, the more it will rearrange into the azlactone L-3, always in equilibrium with 3 through the keto-enol tautomerism (Figure 4)… “   From what I could see in Figure 2, one needs a tertiary base only to deprotonate aminoacid and in order to avoid deprotonation of a-proton in the side-product azlacton L-3, you need to use a base as weak as possible. To me, It fits perfectly with the results shown by pyridine, dimethylaniline, and lutidine (Table 5). My guess is that weak inorganic bases such as sodium bicarbonate or potassium dihydrogen phosphate will also work well in this reaction.   In other words, to me it’s not the base that activate somehow the amino acid for the following racemization, it’s the nature of intermediate HO-Bt-N-Ac-L-phe ester which seems prone to formation of azlactone L-3 which in turn under right condition (base strength) yields to a mixture of the corresponding enantiomers.

Round 2

Reviewer 2 Report

Because the authors appropriately responded to my questions, I agree to accept this paper for publication.

Reviewer 3 Report

I’m satisfied with the revised version of the article and think it could be published as it is. As for our comments on use of inorganic base it’s merely a suggestion since authors run reactions in DMF and heterogenic systems aprotic polar solvent-inorganic base are quite common for instance in the palladium cross-coupling and therefore they may also would prove to be useful/fruitful in authors’ chemistry.